# Molecular resolution imaging by post-labeling expansion single-molecule localization microscopy (Ex-SMLM)

Fabian U. Zwettler[1], Sebastian Reinhard[1], Davide Gambarotto [2], Toby D. M. Bell[3], Virginie Hamel [2✉], Paul Guichard [2✉] & Markus Sauer [1✉]

Expansion microscopy (ExM) enables super-resolution fluorescence imaging of physically expanded biological samples with conventional microscopes. By combining ExM with single-molecule localization microscopy (SMLM) it is potentially possible to approach the resolution of electron microscopy. However, current attempts to combine both methods remained challenging because of protein and fluorophore loss during digestion or denaturation, gelation, and the incompatibility of expanded polyelectrolyte hydrogels with photoswitching buffers. Here we show that re-embedding of expanded hydrogels enables *d*STORM imaging of expanded samples and demonstrate that post-labeling ExM resolves the current limitations of super-resolution microscopy. Using microtubules as a reference structure and centrioles, we demonstrate that post-labeling Ex-SMLM preserves ultrastructural details, improves the labeling efficiency and reduces the positional error arising from linking fluorophores into the gel thus paving the way for super-resolution imaging of immunolabeled endogenous proteins with true molecular resolution.

[1] Department of Biotechnology and Biophysics, Biocenter, University of Würzburg, Am Hubland, 97074 Würzburg, Germany. [2] Department of Cell Biology, Sciences III, University of Geneva, Geneva, Switzerland. [3] School of Chemistry, Monash University, Clayton, VIC 3800, Australia. ✉email: virginie.hamel@unige.ch; paul.guichard@unige.ch; m.sauer@uni-wuerzburg.de

By linking a fluorophore or a protein of interest into a dense, cross-linked network of a swellable polyelectrolyte hydrogel, biological specimen can be physically expanded allowing for magnified imaging with subdiffraction-resolution on conventional microscopes. Since its introduction in 2015[1], expansion microscopy (ExM) has shown impressive results including the magnified visualization of pre- or postexpansion labeled proteins and RNAs with fluorescent proteins (FPs), antibodies, and oligonucleotides, respectively, in isolated organelles, cells, pathogens, tissues, and human clinical specimen[2–4]. In addition, various protocols have been developed to anchor proteins or RNA into charged poly-acrylamide hydrogels[5–8]. Using 2.5% (w/w) acrylamide and 8.55% sodium acrylate with 0.15% (w/w) of the cross-linker *N-N'*-methylenebisacrylamide accomplishes a ~4.5x linear expansion of biological specimens. Decreasing the cross-linker concentration usually permits higher gel expansion factors of up to 10x but also increases proportionally the linkage error defined by the affinity reagent, linker and fluorophore and leads to greater gel instability[9]. It is also possible to expand samples in series enabling gel expansion factors of 20x and higher with a demonstration of 53x expansion of microtubules[10]. However, fluorescence imaging of such greatly enlarged samples is complicated by the dilution of fluorescent labels and dramatic increase in the physical separation between the fluorophore and its target due to the linkage error. Nevertheless, ExM with lower expansion factors enables confocal diffraction-limited fluorescence imaging with spatial resolutions comparable to that of super-resolution microscopy methods[11,12].

To further enhance the resolution, ExM has been combined with structured illumination microscopy (SIM)[13,14] and stimulated emission depletion (STED) microscopy[2,15]. By careful optimization of the expansion protocol U-ExM demonstrated that even ultrastructural details of multiprotein complexes such as centrioles can be truthfully preserved[2]. Combining ExM with SMLM methods (Ex-SMLM) can then potentially further improve the spatial resolution to enable true molecular resolution and bridge the gap to the electron microscopy regime. However, despite these apparent advantages, attempts to combine ExM with SMLM have remained rare and unoptimized due to several challenges[5,16]. There are two major determinants that control the resolution of SMLM, the localization precision and the localization density[11,12]. The localization precision remains unaltered by sample expansion and therefore allows achieving an improved resolution depending on the expansion factor. The localization density is arguably the more important determinant for SMLM on expanded samples. According to information theory, the required density of fluorescent probes has to be sufficiently high to satisfy the Nyquist–Shannon sampling theorem[17]. At its most basic level, the theorem states that the mean distance between neighboring localized fluorophores (the sampling interval) must be at least twice as fine as the desired resolution. In real samples, however, the relationship between localization density and resolution is far more complex[18]. Empirically, it seems that for a given resolution the distance between neighboring localizations should be significantly less than that indicated by a naive application of the Nyquist limit[19].

These considerations illustrate the challenges Ex-SMLM is confronted with. First, the fluorophore density is considerably diluted in expanded samples[9,10], which often results in unclear views of biological structures and complicates SMLM data interpretation. For example, a 4x expansion in three dimensions effectively lowers the labeling density 64-fold. Second, addition of a thiol-containing phosphate-buffered saline (PBS) photoswitching buffer as required for *d*STORM[20,21] to a swellable polyelectrolyte hydrogel with hydrophilic ionic side groups results in substantial shrinking of the gel in the worst case down to its initial size. Finally, ExM protocols use free-radical polymerization

to form polymers. However, free radicals also have the potential to react with the fluorophores which can irreversibly destroy them[1,3,5]. Therefore, the fluorophore density will be further diluted in ExM protocols that use pre-expansion labeling and consequently reduce the structural resolution. The extent of irreversible fluorophore destruction during gelation varies across fluorophores. Unfortunately, the best suited dyes for *d*STORM, the carbocyanine dyes Cy5 and Alexa Fluor 647[19,20], are almost completely destroyed during gelation[1,3,5] Here, post-expansion labeling approaches (post-labeling ExM) offer acceptable solutions[2,7,8], though they require preservation of protein epitopes during expansion.

An additional benefit of post-labeling ExM is improved epitope accessibility for antibodies and a reduction of the linkage error proportional to the expansion factor compared to pre-labeling ExM[2]. For instance, after 4x expansion, the immunolabeling linkage error of 17.5 nm defined by the primary and secondary antibodies[22], would reduce to 4.4 nm, which is the size of a tubulin monomer[23]. Combining SMLM with post-labeling ExM reduces the linkage error by the expansion factor and could thus enable fluorescence imaging with molecular resolution. Here, we set out to develop post-labeling Ex-SMLM with organic fluorophores with minimal linkage error.

## Results

**Re-embedding of expanded samples enables Ex-SMLM in photoswitching buffer.** A major problem of Ex-SMLM is the shrinking of the expanded hydrogels in photoswitching buffer due to ionic interactions between ions of the buffer and the ionic side groups of the gel. Therefore, we re-embedded expanded charged hydrogels in an uncharged polyacrylamide gel as recently introduced for ExM of RNA[6]. We started using pre-labeling ExM in combination with standard immunostaining using unmodified primary and fluorophore labeled secondary antibodies to realize Ex-SMLM (Supplementary Fig. 1). We used microtubules as reference structure to investigate the expansion factor, spatial resolution, structural distortions, and the labeling density. Microtubules are assembled from αß tubulin hetero-dimers, which stack head-to-tail into polar protofilaments with a periodicity of 8 nm, with ~13 protofilaments associating laterally in parallel to form a hollow, polar cylinder (Fig. 1a)[23,24]. As previously measured by transmission electron microscopy (TEM), microtubules are hollow tubes with an outer diameter of 25 nm and 60 nm, respectively, after immunostaining with primary and secondary antibodies[22]. This results in a linkage error defined by the size of the primary and secondary antibody of 17.5 nm (Fig. 1a).

We used the proExM protocol, in which proteins are directly anchored to the hydrogel using the succinimidyl ester of 6-((acryloyl)amino)hexanoic acid (AcX)[5]. To minimize fluorophore loss during gelation in pre-labeling ExM methods, we used the rhodamine derivative Alexa Fluor 532, which retains ~50% of its pre-gelation brightness after expansion[1,3,5]. To prevent shrinking of the hydrogel upon addition of photoswitching buffer, expanded hydrogels were re-embedded in acrylamide for serial staining of the expanded specimen[6]. Hydrogels were incubated twice in 10% AA, 0.15% *bis*-acrylamide, 0.05% APS, 0.05% TEMED in 5 mM Tris (pH 8.9) for 30 min each and subsequently transferred onto coverslips functionalized with acrydite via glass silanization to minimize lateral drift of expanded samples (Supplementary Fig. 1). After polymerization of the re-embedding gel, hydrogels were immersed in photoswitching buffer containing 100 mM mercaptoethylamine (MEA) in PBS. The expansion factor was determined by comparing the post-expansion and post re-embedding fluorescence images with

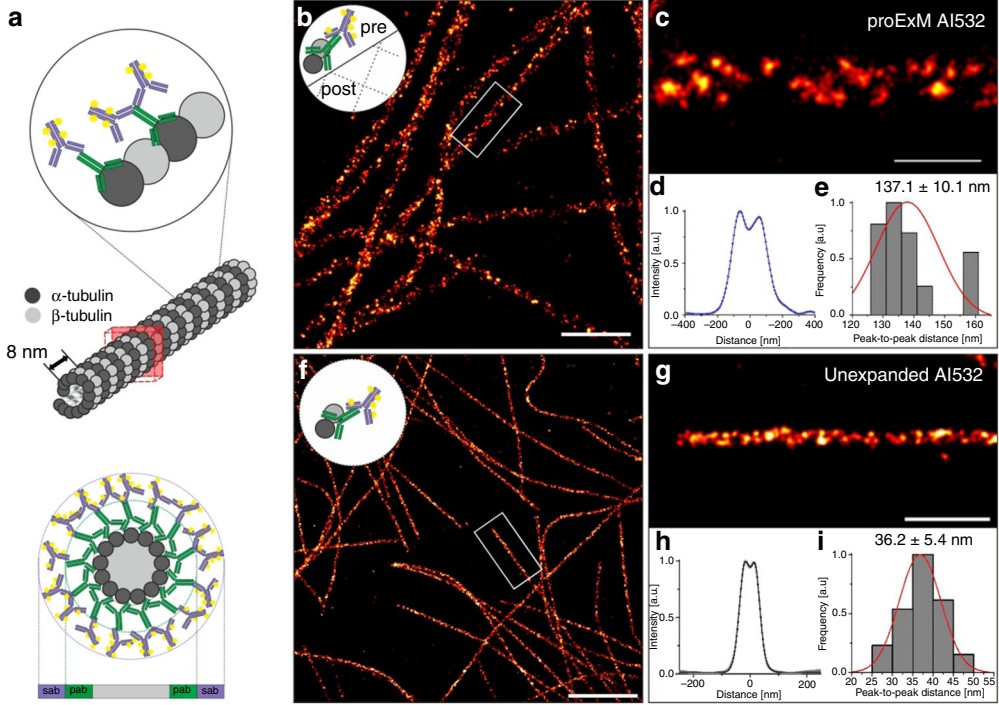

**Fig. 1 Re-embedding enables Ex-*d*STORM. a** Model of microtubules with an outer diameter of 25 nm stained with conventional primary (pab) and fluorescently labeled secondary IgG antibodies (sab) results in a total diameter of 60 nm with a linkage error (defined by the size of the primary and secondary antibody) of 17.5 nm[22]. **b** *d*STORM image of pre-labeled proExM expanded and re-embedded Cos-7 cells stained with primary antibodies against α-tubulin and secondary Alexa Fluor 532 conjugated antibodies (Al532). The small logo in the upper left corner symbolizes that microtubules have been immunolabeled before expansion (pre-labeled). **c** Zoom in on highlighted region in (**b**). **d** Averaged cross-sectional profile of nine microtubule segments with a total length of 29.1 μm (segment lengths range from 2.1-4.5 μm) measured in two cells from 1 expanded sample. **e** Histogram of peak-to-peak distances with normalized normal distribution curve (red) determined by bi-Gaussian fitting of the data analyzed in (**c**) with an average distance of 137.1 ± 10.1 nm (mean ± sd). The data were obtained from n = 9 microtubule segments in 2 cells from 1 expanded sample. **f** Unexpanded Cos-7 cells labeled with an anti α-tubulin primary antibody and Alexa Fluor 532 (Al532) conjugated IgG secondary antibodies. The small logo in the upper left corner symbolizes that microtubules have been immunolabeled and not expanded. **g** Zoom in of the white boxed region in (**f**). **h** Average intensity profile of 35 microtubule segments with a length between 1.1 and 5.8 μm (mean = 2.0 μm) and a total length of 69.6 μm analyzed in 12 *d*STORM images. **i** Histogram of peak-to-peak distances with normalized normal distribution curve (red) determined by bi-Gaussian fitting of cross-sectional profiles of the analyzed microtubule segments in (**h**) with a mean peak-to-peak distance of 36.2 ± 5.4 nm (mean ± sd). The data were obtained from n=35 microtubule segments in 12 cells and 3 independent experiments. Scale bars, 2 μm (**b**, **f**), 500 nm (**c**, **g**).

pre-expansion fluorescence images. The results showed a low distortion introduced by the re-embedding process and a reduction in gel size of ~20% from ~3.9x before to ~3.1x after re-embedding (Supplementary Figs. 2 and 3).

A caveat of imaging expanded samples is that super-resolution imaging methods, and in particular SMLM, are most effective when used on thin samples located within a few micrometers above the coverslip surface. However, expanded specimen can be easily located several tens of micrometers above the coverslip. In addition, expanded specimens are transparent because they consist largely of water. Hence, the use of oil-immersion objectives and total internal reflection fluorescence (TIRF) microscopy as used in most SMLM applications to achieve a higher signal-to-background ratio is in this case not the best choice. Therefore, we decided to use a water-immersion objective and epi-illumination in all SMLM experiments. The corresponding *d*STORM images of pre-labeled expanded microtubules showed homogeneously labeled filaments with some labeling gaps reflecting fluorophore and protein loss during polymerization and enzymatic digestion, respectively (Fig. 1b, c). In addition, we imaged unexpanded microtubules by *d*STORM under identical experimental conditions (Fig. 1f, g).

To examine the achieved spatial resolution, cross-sectional profiles of selected microtubule areas are often consulted[21]. If the

two-dimensional (2D) projection of the fluorescence intensity distribution measured from microtubule filaments show a bimodal distribution, the peak-to-peak distance can then be fitted with a sum of two Gaussians and used as an estimate of the spatial resolution. To ensure an objective evaluation and comparison of the spatial resolution achieved, we developed "Line Profiler", an automated image processing software. Line Profiler automatically evaluates potential regions of interest along filamentous structures to generate cross-sectional profiles that can be fit by a sum of two Gaussians to determine the peak-to-peak distance between the sidewalls of the filamentous structure (Supplementary Fig. 4).

In order to compare the experimentally measured peak-to-peak distances of different expansion protocols, we simulated trans-verse profiles of unexpanded and expanded microtubules using a cylindrical distribution function to describe the hollow annular structure of microtubules (Fig. 2a and Supplementary Fig. 5) similar to the approach used recently for iterative expansion[10]. The resulting peak-to-peak distances were used to determine the molecular expansion factor of expanded immunolabeled microtubules considering the influence of the label size on the broadening of the microtubule diameter (Supplementary Table 1 and Supplementary Note 1).

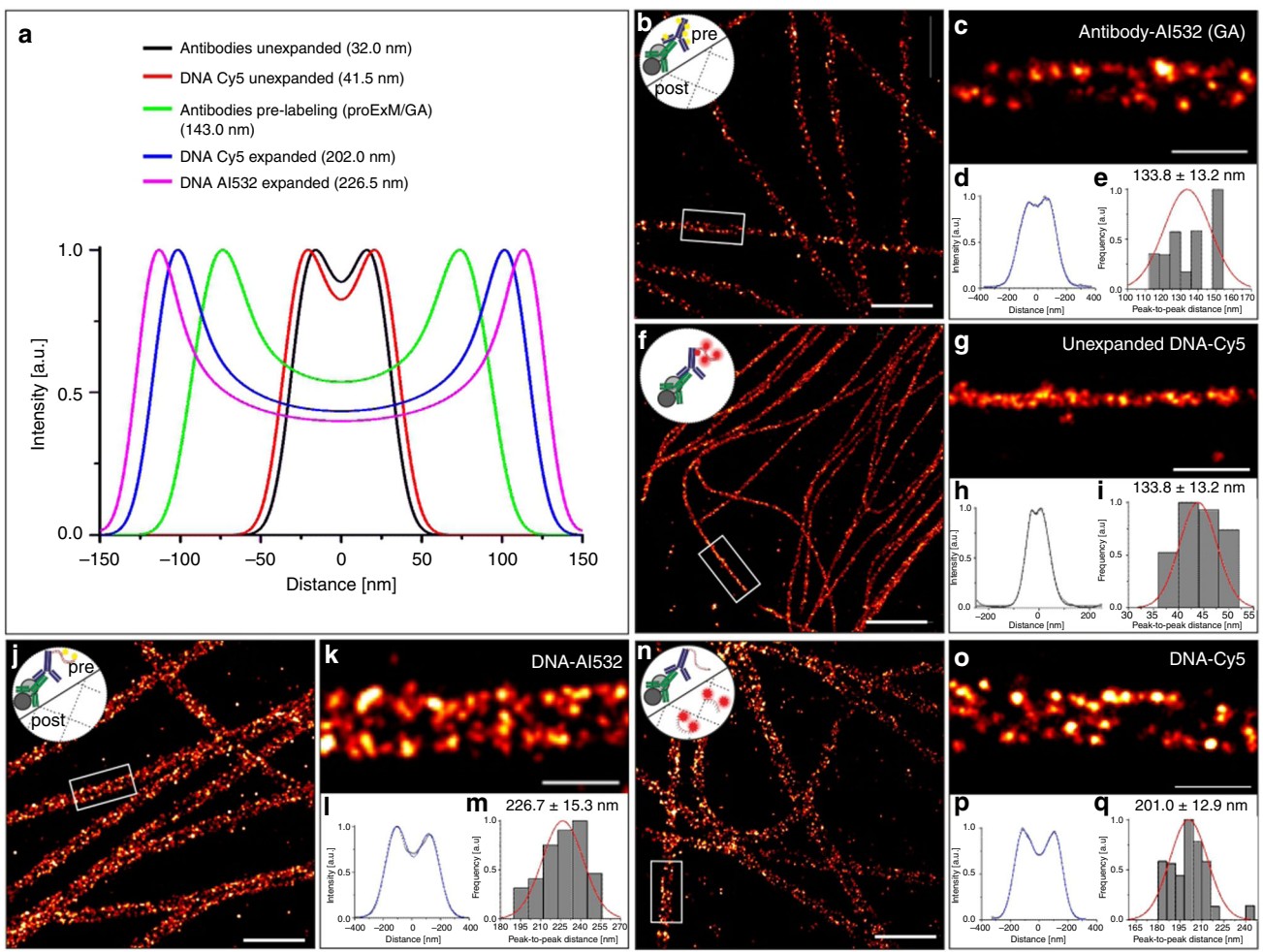

**Fig. 2 Pre-labeling Ex-*d*STORM. a** Simulated intensity profiles using a cylindrical distribution function to describe unexpanded or 3.2x expanded immunostained microtubules (labeled with IgG antibodies or DNA modified IgG antibodies pre-expansion) and resulting peak-to-peak distances of the cross-sectional profiles. **b** *d*STORM image of expanded and re-embedded α- and β-tubulin pre-labeled with secondary Alexa Fluor 532 IgG antibodies (Al532) using the MA-NHS/GA method[6], i.e. antibodies are cross-linked with glutaraldehyde (GA) into the hydrogel (Antibody-Al532 (GA)). **c** Zoom in of white boxed region in (**b**). **d** Averaged cross-sectional profile of 8 microtubule segments with a length between 1.5–6.4 μm and 28.6 μm in total measured in 4 expanded cells. **e** Histogram of peak-to-peak distance distribution with normalized normal curve (red) of microtubule segments analyzed in (**d**) at $n =$ 8 microtubule segments in 4 cells from 1 expansion experiment with a mean distance of 133.8 ± 13.2 nm (mean ± sd). **f** Unexpanded *d*STORM image of ssDNA-Cy5 secondary antibody hybridized with Cy5 bearing oligonucleotides pre-expansion (DNA-Cy5 protocol). **g** Magnified view of white boxed region in (**f**). **h** Average cross-sectional profile of 7 microtubule segments with a length between 1.0–1.8 μm and 8.7 μm in total. **i** Histogram of peak-to-peak distances with normalized normal distribution curve (red) of the data analyzed in (**h**) along $n = 7$ microtubule segments in 2 cells from 1 experiment with a mean distance of 43.9 ± 3.7 nm (mean ± sd). **j** Expanded *d*STORM image of microtubules labeled with α-tubulin and dsDNA (DNA-Al532) conjugated secondary antibodies exhibiting a methacryloyl group to crosslink the DNA with fluorophores pre-expansion into the hydrogel (original ExM trifunctional label concept)[1]. **k** Zoom-in of white boxed region in (**j**). **l** Average intensity profile of 26 microtubule segments with a length of 2.4–10.7 μm and 118.6 μm in total. **m** Histogram of peak-to-peak distances with normalized normal distribution curve (red) determined from $n = 26$ microtubule segments in 4 cells from 1 expanded sample showing a mean distance of 226.7 ± 15.3 nm (mean ± sd). **n** *d*STORM image of α- and β-tubulin expanded according to the DNA-Cy5 protocol strategy with labels at Cy5-bearing oligonucleotides introduced post-re-embedding. **o** Zoom in of white boxed region in (**n**). **p** Average intensity profile of 15 microtubule segments with a length between 1.6–25.1 μm and a total length of 126.0 μm in 1 expanded sample. **q** Histogram of peak-to-peak distances with normalized normal distribution curve (red) determined by fitting the cross-sectional profiles analyzed in (**p**) along $n = 22$ microtubule segments in 4 cells from 1 expanded sample showing a mean distance of 201.0 ± 12.9 nm (mean ± sd). The small logos in the upper left corner symbolize the labeling method, e.g. pre- and post-immunolabeled with or without DNA-linker, respectively. Scale bars, 2 μm (**b**, **f**, **j**, **n**), 500 nm (**c**, **g**, **k**, **o**).

**Pre-labeling Ex-SMLM**. *d*STORM images of unexpanded and expanded microtubules clearly showed a bimodal signal distribution along the filaments, similar to that of previous super-resolution microscopy studies (Fig. 1c, d and 1g, h)[21,25]. When the cross-sectional profile of unexpanded microtubules was fit with a sum of two Gaussians, the peak-to-peak distance between the sidewalls showed a mean value of 36.2 ± 5.4 nm (mean ± s.d.) analyzed over several microtubule filament segments (Fig. 1i). This value is

expected for the projection of a 25 nm inner diameter cylinder that has been broadened by primary and secondary antibodies on both sides by 17.5 nm[22] (Fig. 1a) and corresponds well to the simulated peak-to-peak distance of 32.0 nm for unexpanded microtubules (Fig. 2a and Supplementary Note 1). The mean peak-to-peak distance of proExM treated and expanded microtubules was determined to 137.1 ± 10.1 μm (mean ± s.d.) (Fig. 1e). This value corresponds to an expansion factor of 3.1x determined from

simulation of expanded microtubules pre-labeled with primary and secondary IgG antibodies (Supplementary Table 1).

We next used the post-stain linker-group functionalization method (MA-NHS/GA method)[7] as an alternative pre-labeling Ex-SMLM method. Here, the entire sample was functionalized with polymer-linking groups after conventional immunostaining. The resulting dSTORM images showed a peak-to-peak distance between the microtubule sidewalls of 133.8 ± 13.2 nm (mean ± s. d.) (Fig. 2b–e) corresponding to a simulated expansion factor of 3.0x (Supplementary Table 1). The determined peak-to-peak distance is in good agreement with the peak-to-peak distance determined from proExM expanded microtubules (Fig. 1b–e). Variations in the measured peak-to-peak distances can be well explained by varying initial expansion factors of hydrogels which are typically in the range of ~4.0–4.5x for the used ExM gel composition. Considering a ~20% reduction in gel size caused by re-embedding of the hydrogel, an ultimate expansion factor of ~3.1–3.6x can be expected which fits well with the determined molecular expansion factors.

Next, we tested the original ExM protocol with trifunctional DNA-modified secondary antibodies[1], which can be labeled with dye-functionalized complementary oligonucleotides that contain an acrydite linker modification. Alternatively, antibodies can be modified with a single stranded DNA that is incorporated directly into the hydrogel. Antisense dye-labeled oligonucleotides can then be hybridized after re-embedding of the hydrogel, which enables the use of fluorophores that would not survive the radical polymerization process. Since the linkage error is mainly determined by the IgG antibodies and the 40 bases long DNA strand (Supplementary Table 2) both methods still belong to the pre-labeling Ex-SMLM method (Supplementary Fig. 1).

First, we tested the approach on unexpanded microtubules and obtained peak-to-peak distances of 43.9 ± 3.7 nm (mean ± s.d.) (Fig. 2f–i) and 37.0 ± 4.8 nm (mean ± s.d.) (Supplementary Fig. 6) for labeling with Cy5- and Alexa Fluor 532-modified oligonucleotides, respectively. These values are in good agreement with the theoretically expected value of 41.5 nm for immunolabeling with 42 bases long trifunctional oligonucleotide-modified secondary antibodies (Fig. 2a). Due to the additional modification of the secondary antibodies, the peak-to-peak distances should be a few nanometers larger than the value measured for standard immunolabeled microtubules of 36.2 ± 5.4 nm (mean ± s.d.) (Fig. 1f–i).

If the oligonucleotide-modified secondary antibodies are labeled with complementary Alexa Fluor 532-modified oligonucleotides prior to expansion, we measured a peak-to-peak distance of 226.7 ± 15.3 nm (mean ± s.d.) from dSTORM images (Fig. 2j–m). Since Cy5 does not survive gelation[1,3,5], we performed an additional experiment labeling the oligonucleotide-modified secondary antibodies after expansion with complementary Cy5-modified oligonucleotides, performed dSTORM imaging in photoswitching buffer and determined a slightly shorter peak-to-peak distance of 201.0 ± 9.3 nm (mean ± s.d.) (Fig. 2n–q). Both values are in excellent agreement with the theoretical peak-to-peak distance of 226.5 nm and 202 nm, respectively (Fig. 2a), simulated for 3.2x expanded microtubules taking into account the length of the 42 base pair trifunctional oligonucleotide, the position of fluorophores within the DNA strand and its spatial orientation (Supplementary Fig. 7 and Supplementary Note 1). The slightly shorter peak-to-peak distance measured in the Cy5-experiment where the dye-labeled complementary strand was hybridized after expansion can be explained most likely by coiling of the single-stranded trifunctional oligonucleotide during gelation (Supplementary Fig. 7). These results indicate that Ex-SMLM can resolve linker differences of 42 DNA base pairs (corresponding to ~14.3 nm) and, interestingly conformational differences between single and double-stranded DNA linkers.

Noteworthy is that the total size of an expanded sample is not only determined by the biomolecule of interest, e.g. microtubules, but also by the fluorescent probe, e.g. primary and secondary antibodies, used to label the biomolecule of interest. Unexpanded, microtubules labeled with primary and secondary IgG antibodies exhibit a diameter of ~60 nm with a linkage error (defined by the size of the primary and secondary antibody) of 17.5 nm[22]. For example, for 3.3x expansion this translates into a microtubule diameter of 3.3 × 25 nm = 82.5 nm whereas the diameter of the immunolabeled microtubule is substantially broadened to ~198 nm because of the linkage error of 3.3 × 17.5 nm = 57.75 nm introduced by the primary and secondary antibody that has to be added to both sides of the microtubule (Supplementary Fig. 5). In other words, even though SMLM achieves high localization precisions[12,13], a linkage error of > 50 nm undoes much, or even all, of the gain in resolution.

**Pre- versus post-labeling Ex-SMLM**. In order to reduce the linkage error, we next explored post-labeling Ex-SMLM. It has been shown that the fluorescence signals from some genetically encoded FPs as well as conventional fluorescently labeled secondary antibodies and streptavidin that are directly anchored to the gel are at least partially preserved by proExM even when subjected to the strong nonspecific proteolytic digestion used in the original ExM protocol[1,5]. Therefore, we anticipated that protein epitopes might survive the proExM protocol[26]. To compare the labeling density of pre- and post-labeling Ex-SMLM we immunostained microtubules with Alexa Fluor 532 before and additionally after expansion. Intriguingly, combining pre- with post-labeling resulted in a substantial shortening of the average peak-to-peak distance of the sidewalls of microtubules to 79.5 ± 6.6 nm (mean ± s.d.) determined from dSTORM images (Fig. 3). We speculated that the protease digestion step may destroy a large fraction of the pre-labeled antibody complexes but to our surprise, the majority of tubulin epitopes survives this critical step. Together with the increased accessibility of tubulin epitopes for primary antibodies and primary antibody epitopes for secondary antibodies after expansion this results in peak-to-peak distances undistinguishable from solely post-labeled microtubules.

To examine more quantitatively epitope survival and increased epitope accessibilities, we simulated the cross-sectional profiles expected for pre- and post-labeled microtubules. Here we assumed a ~10-fold signal dilution ($3.2^2$) for the 2D projection of the fluorescence signals of 3.2x expanded pre-labeled antibodies (Fig. 3e and Supplementary Fig. 5). Hence, the cross-sectional microtubule profiles show the superposition of the profile calculated for the 3.2x expansion of 25 nm diameter microtubules post-immunolabeled and 60 nm diameter microtubules pre-immunolabeled. The resulting superposition profile shows a peak-to-peak distance of 79.5 nm (Fig. 3e) emphasizing the advantage of post-labeling Ex-SMLM. Post-labeling Ex-SMLM using the proExM protocol[5] provides an improved labeling efficiency and a reduced linkage error. In fact, the immunolabeling linkage error of ~58 nm for pre-labeling reduces to ~5 nm for post-labeling considering a 3.2x expansion factor and thus improves the effective achievable resolution (Supplementary Fig. 8).

Therefore, dSTORM images of Alexa Fluor 532 labeled microtubules clearly revealed the hollow cylinder of microtubules (Fig. 3c) using a water-immersion objective and epi-illumination, similar to recently published results obtained by DNA-PAINT TIRF microscopy and experimental point spread function fitting[27]. The average distance between the sidewalls of

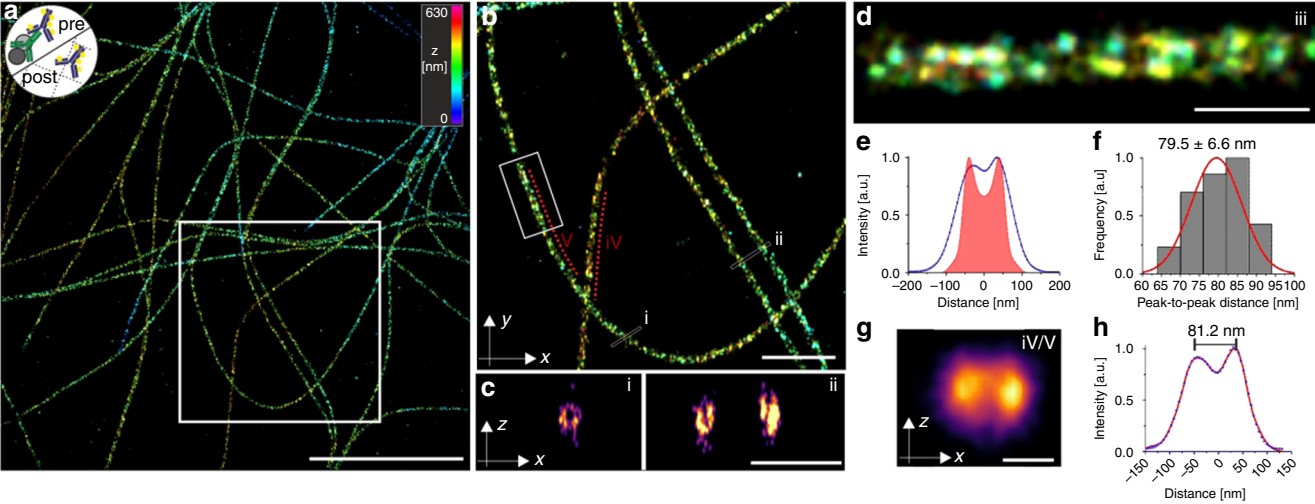

**Fig. 3 3D post-labeling Ex-*d*STORM.** SMLM image of re-embedded and post-expansion labeled microtubules. **a** 3D *d*STORM image of re-embedded Cos-7 cells expanded according to the Protein-Retention protocol (proExM)[4] pre-labeled with anti α- and β-tubulin antibodies and additionally post-labeled with anti α-tubulin. The secondary antibodies were labeled with Alexa Fluor 532 (proExM Al532). The small logo in the upper left corner symbolizes the labeling method, e.g. pre- and post-immunolabeling with Al532 secondary antibodies. **b** Magnified view of highlighted region in (**a**). **c** *xz*-side view cross-sections (white lines) (i) and (ii) shown in (**b**) revealing the hollow structure of microtubules. **d** Magnified view of highlighted region (white box) in (**b**). Since post-labeling dominates the signal, the method is termed proExM Al532 (post-labeled). **e** Averaged cross-sectional profile (blue) of 11 analyzed microtubule segments along a total of 28.2 μm filament (2.1–5.5 μm segments) of one expanded sample. The simulated cross-sectional profile for 3.2x proExM expanded pre- and post-labeled microtubule assuming a pre- to post-labeling ratio of 0.1 is shown in red. **f** Histogram of peak-to-peak distances with normalized normal curve (red) of fitted profiles analyzed in (**e**) with an average distance of 79.5 ± 6.6 nm (mean ± sd) analyzed along *n* = 11 microtubule segments in 2 cells from 1 expanded sample. **g** Image projection of the *xz*-axes averaged along two microtubule filaments (iv) and (v) shown in (**b**) (red dotted lines) using the "z projection analysis" of the software "Line Profiler". **h** Cross-sectional profile (blue dots) of the *xz*-projection shown in (**g**). Using a bi-Gaussian fit (red) the peak-to-peak distance is determined to 81.2 nm. Scale bars, 10 μm (**a**), 5 μm (**b**), 1 μm (**c**), 500 nm (**d**), 100 nm (**g**).

the *xz*-projection of a 6.5 μm long microtubule filament was determined to 81.2 nm (Fig. 3g–h) highlighting the high spatial resolution of pre-labeling 3D Ex-*d*STORM.

**Post-labeling Ex-SMLM of centrioles.** Motivated by the results, we set out to explore the molecular organization of centriole organelles by Ex-SMLM. We used isolated *Chlamydomonas* centrioles, which have a characteristic 9-fold microtubule triplet-based symmetry, forming a polarized cylinder ~ 500 nm long and ~ 220 nm wide[28] (Supplementary Fig. 9). Recently[2], U-ExM has been developed as an extension of ExM that allows for near-native expansion of organelles and multiprotein complexes and visualization of preserved ultrastructures by optical microscopy. When combined with STED microscopy, details of the ultrastructural organization of isolated centrioles such as the 9-fold symmetry and centriolar chirality could be visualized[2]. Advantageously, U-ExM uses post-labeling to improve the epitope accessibility after expansion. Here, we used U-ExM treated centrioles in combination with post-labeling with Alexa Fluor 647 secondary antibodies to enable *d*STORM imaging, which has previously be impossible due to shrinking of expanded hydrogels in photoswitching buffer. Therefore, samples were re-embedded and transferred onto coverslips functionalized with acrydite via glass silanization to minimize lateral drift. This allowed us to perform post-labeling 3D Ex-*d*STORM on ~3.4x expanded centrioles (Fig. 4a–d and Supplementary Fig. 10).

Alternatively, we used the spontaneously blinking Si-rhodamine dye HMSiR[29] that enables SMLM in the absence of photoswitching buffer and does thus not require re-embedding. Using double-deionized water, we achieved a molecular expansion factor of ~4x (Fig. 4d–f and Supplementary Fig. 9). Unfortunately, since the pH of double-deionized water is below 7.0, HMSiR does not exhibit optimal blinking characteristics[29]. Addition of PBS buffer, pH 7.4

improved the blinking characteristics of HMSiR but reduced the expansion factor to ~2x, which limits the spatial resolution of the SMLM experiments (Fig. 4d, g). In contrast to 3D *d*STORM images of unexpanded centrioles (Fig. 4h) post-labeling 3D Ex-SMLM clearly visualized the centriole as a bundle of nine microtubule triplets. SMLM images of expanded isolated *Chlamydomonas* centrioles showed the 9-fold symmetry of the procentrioles (Fig. 4b, f) with tubulin diameters of ~220 nm in agreement with previous studies[2,30]. Even in side views of centrioles imaged by 3D Ex-*d*STORM the neighboring microtubule triplets are clearly separated (Fig. 4c). Furthermore, 3D Ex-*d*STORM allowed us to resolve ring-like sub-structures of centrioles indicating hollow cylinders of microtubule triplets (Supplementary Fig. 11). According to these results, re-embedding of the sample and *d*STORM in photoswitching buffer provides currently the best Ex-SMLM performance. Since microtubule triplets separated by 15–20 nm[30] are very well resolved in the expanded images post-labeling Ex-SMLM exhibits a spatial resolution that is way below 15–20 nm reaching the structural resolution required to resolve the molecular architecture of centrioles.

**Discussion**

Electron microscopy has been the only viable method able to reveal the ultrastructure of organelles and molecular complexes for decades because of the diffraction limit of optical microscopy. Super-resolution microscopy offers up to ~10x higher resolution than conventional diffraction-limited microscopy[11,12]. Improved super-resolution microscopy methods can now localize single emitters with a precision of a few nanometers[31–33], but limitations in labeling efficiency and linkage error have thus far prevented the translation of high localization precision into sub-10-nm spatial resolution. Therefore, the spatial resolution provided by all these inventive methods is currently still too low

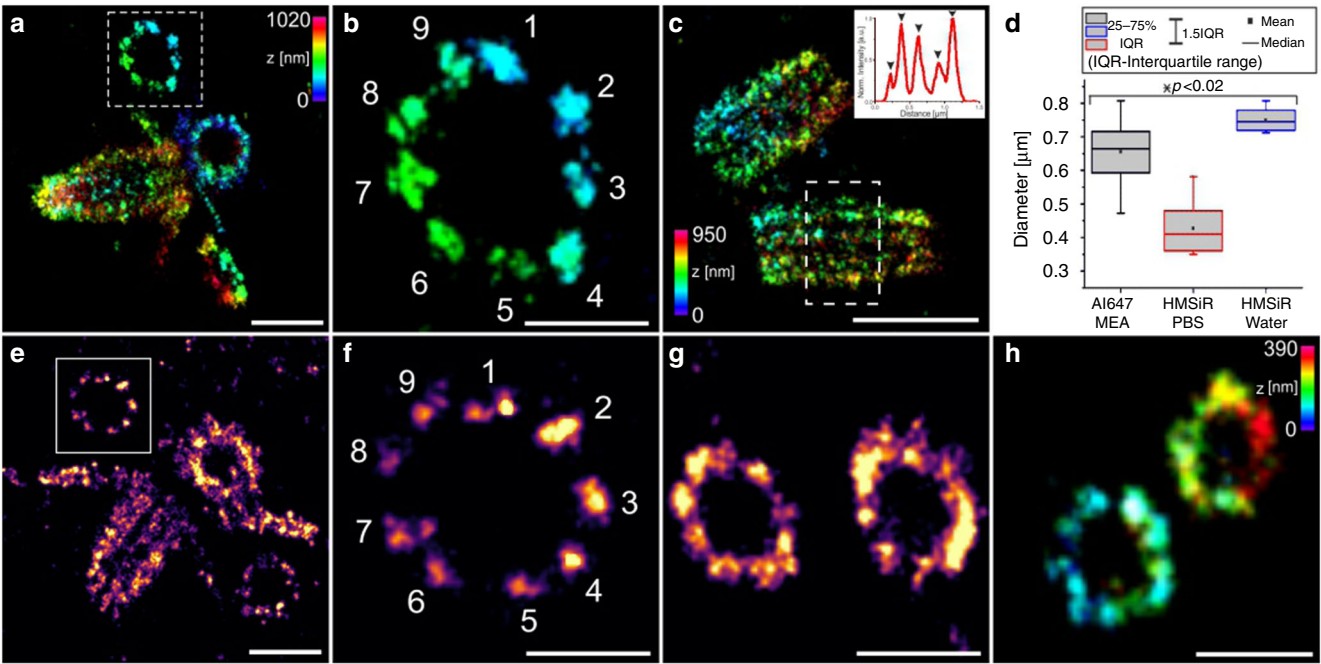

**Fig. 4 Ex-SMLM of U-ExM expanded centrioles. a-c** 3D *d*STORM image of U-ExM expanded and re-embedded *Chlamydomonas* centrioles stained post re-embedding with anti α-tubulin primary antibody and Alexa Fluor 647 conjugated secondary antibodies measured in MEA buffer. **b** Zoom-in on highlighted region in (**a**) revealing the 9-fold symmetry of the shown procentriole. **c** Side view of two mature centrioles with clearly separated triplets. The inlet shows the cross-sectional profile along the centriole (white box) showing five distinct peaks of microtubule triples (marked with arrow heads). **d** Comparison of the diameters determined from expanded centrioles measured using different protocols (re-embedded and labeled with Alexa Fluor 647, and imaged in MEA photoswitching buffer, labeled with HMSiR 647 and imaged in double-deionized water or in pH optimized PBS (1x) buffer with pH 7.4). Mean values are 657 ± 90 nm (mean ± sd) for Alexa Fluor 647 in MEA buffer (*n* = 12 centrioles), 428 ± 74 nm (mean ± sd) for HMSiR in PBS (*n* = 7 centrioles), and 750 ± 34 nm (mean ± sd) for HMSiR 647 in water (*n* = 8 centrioles). Data from *n* = 2 independent experiments for each condition. Divided by the previously analyzed diameter of α-tubulin labeled centriole expansion factors translates into ~3.4x, ~2.2x, and ~3.9x for expanded centrioles labeled with Alexa Fluor 647 in MEA buffer, HMSiR in PBS (1x), and HMSiR 647 in water, respectively. Statistical significance was assessed by one-way ANOVA: the mean values of the diameters are significantly different with $p<0.02$ (F = 3.80) (*$P \leq 0.05$, **$P \leq 0.01$). **e-g** 2D *d*STORM image of U-ExM expanded centrioles labeled with HMSiR 647 imaged in water (**e-f**) or PBS(1x) (**g**) **f** Zoom-in on highlighted region in (**e**). **h** 3D *d*STORM image of unexpanded isolated *Chlamydomonas* centrioles immunostained with antibodies against glutamylated tubulin and Alexa Fluor 647 conjugated secondary antibodies. Scale bars, 1 μm (**a**, **e**), 500 nm (**b**, **f**, **g**), 1.5 μm (**c**), 250 nm (**h**).

to unravel the composition and molecular architecture of protein complexes or dense protein networks. Expansion microscopy (ExM) represents an alternative approach to bypass the diffraction barrier. By linking a protein of interest into a cross-linked network of a swellable polyelectrolyte hydrogel, biological specimens can be physically expanded allowing for sub-diffraction resolution imaging on conventional microscopes[1–10]. However, even in combination with super-resolution microscopy techniques, spatial resolutions below ~20 nm have so far proven to be very difficult to achieve by ExM[16]. Here, we have shown that re-embedding of expanded hydrogels enables the use of standard photoswitching buffers and *d*STORM imaging of ~3.2x expanded samples. Our results demonstrate that post-labeling ExM using the proExM protocol[5] or U-ExM[2] provides solutions for the two major limiting problems of improved super-resolution microscopy, the labeling efficiency and linkage error. First, as shown for microtubules, expansion of the sample increases the epitope accessibility and thus the labeling efficiency. Comparison experiments demonstrated that post-labeling outperforms pre-labeling several times in this regard (Fig. 3). Second, post-labeling ExM reduces the linkage error proportionally to the expansion factor. Hence, post-immunolabeling of 3.2x expanded microtubules reduces the linkage error from 17.5 nm[22] to ~5 nm (Fig. 3). Since the linkage error also influences the localization accuracy and thus the effective achievable resolution (Supplementary Figs. 8 and 9)[34,35] our findings are highly relevant. Very

recently[36,37], trifunctional linkers have been introduced that are inert to polymerization, digestion and denaturation, and enable direct covalent linking of target molecules and functional groups to the hydrogel. Therefore, trifunctional linkers can retain a high number of labels and fluorescence molecules available for post-expansion imaging. However, since the target molecules are labeled with primary and secondary antibodies or enzymatic tags (e.g. SNAP-tags) functionalized with the trifunctional anchor before expansion, linkage errors remain. The improved labeling efficiency of post-labeling Ex-*d*STORM in combination with small (1.5 × 2.5 nm) camelid antibodies ("nanobodies")[38,39] and 10−20x expansion factors[9,10] can thus pave the way for true molecular resolution imaging of endogenous proteins with 1–5 nm spatial resolution. On the other hand, at such a small length scale, distortions of the structure may occur. To realize more homogeneous gel network structures, a new gelation method based on a highly homogeneous expansion microscopy polymer composed of tetrahedron-like monomers has been introduced[40]. The new tetra-gel polymer chemistry may introduce fewer spatial errors than earlier versions, and enable molecular resolution post-labeling Ex-*d*STORM with reduced distortion. Nevertheless, already ~3x Ex-SMLM can resolve small linker length and conformational differences between labeling approaches as shown here for oligonucleotide-functionalized secondary antibodies (Fig. 2). In addition, we have shown that post-labeling 3D Ex-*d*STORM exhibits excellent structure preservation and already

3.4x expansion using standard protocols can provide a sufficient structural resolution to resolve details of the molecular architecture of centrioles (Fig. 4 and Supplementary Fig. 11).

## Methods

**Reagents**. Acetic acid (A6283, Sigma), Acrylamide (AA, 40%, A4058, Sigma), Acryloyl-X, SE, 6-((acryloyl)amino)hexanoic Acid, Succinimidyl Ester (A20770, Thermo Fisher), Agarose (A9539, Sigma), Ammonium persulfate (APS, A3678, Sigma), Bind Silane (GE17-13330-01, Sigma), Bovine Serum Albumin (BSA, A2153, Sigma), Cysteamine hydrochloride (MEA, 6500, Sigma), Dextran sulfate (D8906, Sigma), DMEM/HAM's F12 with L-glutamine (Sigma, D8062), Ethanol (absolute, ≥ 99.8%, 32205, Sigma), Ethylenediaminetetraacetic acid (E1644, Sigma),

Ethylene glycol-bis(2-aminoethylether)-N,N,N′,N′-tetraacetic acid (EGTA, 03777, Sigma), 10% FBS (Sigma, F7524), Formaldehyde (FA, 36.5-38%, F8775, Sigma), Glutaraldehyde (GA, 25%, G5882, Sigma), Guanidine hydrochloride (50933, Sigma), 2-(N-morpholino)ethanesulfonic acid (MES, M3671, Sigma), N,N′-methylen-bisacrylamide (BIS, 2%, M1533, Sigma), N,N,N′,N′-Tetramethylethylenediamine (TEMED, T7024, Sigma), Poly-D-lysine hydropromide (P6407, Sigma), Polyoxyethylene (20) sorbitan monolaurate solution (Tween-20, 10%, 93774, Sigma), Potassium hydroxide (P5958, Sigma).

Proteinase K (P4850, Sigma), Saline sodium citrate buffer (SSC,20x, 15557, Thermo Fisher), Sodium acrylate (SA, 97-99%, 408220, Sigma), Sodium chloride (NaCl, S7653, Sigma), Sodium dodecyl sulfate (SDS, L3771, Sigma), streptomycin (0.1 mg/ml) (Sigma, R8758), Tris base (T6066, Sigma), Triton X-100 Surfact-Amps Detergent Solution (10% (w/v), 28313, Thermo Fisher), Yeast tRNA (AM7119, Thermo Fisher).

**Antibodies and labeling reagents**. Rabbit anti α-tubulin antibody (ab18251, abcam), Mouse anti β-tubulin antibody (T8328, Sigma), Mouse anti poly-glutamylated tubulin, mAb (GT335) (Adipogen, AG-20B-0020), Alexa Fluor 647 F (ab')2 of goat anti rabbit IgG (A-21246, Thermo Fisher), Alexa Fluor 647 F(ab')2 of goat anti mouse IgG (A-21235, Thermo Fisher, Al532-Goat anti Rabbit IgG (H+L) (A-11009, Thermo Fisher), Al532-Goat anti Rabbit IgG (H +L) (A-11002, Thermo Fisher), HMSiR 647 (A208-01, MoBiTec) conjugated to goat anti rabbit IgG F(ab')2 (SAB3700946, Sigma), TetraSpeck Microspheres (0.1 μm, T27279, Thermo Fisher).

***Chlamydomonas reinhardtii* centriole isolation**. Centrioles were isolated from the cell wall-less *Chlamydomonas reinhardtii* strain CW15 by centrifugation at 600g for 10 min in 50 ml conical tubes[41]. Isolated centrioles were thawed on ice and diluted with cold K-Pipes 10 mM pH 7.2. Centrioles were then loaded in a 15 ml Corex tube with a homemade adaptor and concentrator, and spun onto a 12 mm Poly-D-lysine coated coverslip through centrifugation at 10,000g for 10 min with a JS-13.1 swinging bucket rotor (Beckman) at 4 °C. Coverslips were then processed for immunostaining and expansion microscopy.

**Cell culture of mammalian cells**. COS-7 monkey kidney cells (purchased from CLS Cell Line Servie GmbH) were cultured at 37 °C and 5% CO2 in DMEM/HAM's F12 medium with L-glutamine containing FBS (10%) and penicillin (100 U/ml) and streptomycin (0.1 mg/ml). 20–30,000 cells per well were seeded on round 18 mm high precision cover glasses (No 1.5) in 12-well culture plates (Techno Plastic Products, 92012) and grown for 24 h prior to fixation.

**Sample preparation**. For fixation, all solutions were pre-warmed to 37 °C and fixation was conducted on a heating plate set to 37 °C. Right before fixation samples were rinsed once with pre-warmed Cytoskeleton buffer (CB-buffer, 10 mM MES, 150 mM NaCl, 5 mM EGTA, 5 mM glucose and 5 mM MgCl2, pH 6.1). Cells were then fixed and permeabilized simultaneously incubating a primary fixative solution of 0.3% glutaraldehyde and 0.25% Triton X-100 in CB-buffer for 90 s followed by a second fixation using 2% glutaraldehyde in CB-buffer for 10 min. Fixation was stopped by a 7 min reduction step with freshly prepared 0.5% NaBH4 in PBS. Specimen were then washed three times with PBS (1x) for 5 min each and treated differently depending on subsequent expansion method described below.

Unless otherwise stated all incubations were carried out at room temperature in the following protocols. Immunostaining was either performed pre-gelation (referred to as pre-labeling), post-expansion (post-labeling) or post-re-embedding (post re-embedding labeling). Sequences and modifications of DNA labels are listed in Supplementary Table 2. A list of primary and secondary antibodies used for immunostaining in the corresponding Figures is provided in Supplementary Table 3 and Supplementary Table 4 with details about the expansion protocol used.

**Immunostaining of unexpanded Cos-7 cells**. Cells were placed in blocking buffer (5% BSA in PBS) for 1 h and then incubated for 1 h with anti-alpha tubulin primary antibody solution (ab1825, diluted 1:500, final concentration $c_{end} = 2$ μg/ml) diluted in blocking buffer. Samples were washed thrice in PBS (1x) for 5 min each and incubated with secondary Alexa Fluor 532 IgG antibody solution in blocking buffer (A-11002, diluted 1:200, $c_{end} = 10$ μg/ml) for 1 h followed by three washes in PBS (1x) for 10 min each.

**ExM protocol using DNA trifunctional labels (ExM protocol)**. After blocking with 5% BSA in PBS for 1 h, cells were incubated with anti-alpha tubulin primary antibody (ab1825, diluted 1:500, $c_{end} = 2$ μg/ml) in blocking buffer (5% BSA in PBS) for 1 h, followed by three washes in PBS (1x) for 5 min each and incubation of "Antibody B" DNA-labeled secondary antibodies (10 μg/ml) in hybridization buffer (2x saline sodium citrate (SSC), 10% dextran sulfate, 1 mg/ml yeast tRNA, 5% BSA) for 3 h. Antisense DNA B1-Alexa Fluor 532 and DNA B2-Alexa Fluor 532 oligos were hybridized simultaneously at a total DNA concentration of 1.0 ng/μl for 3 h in hybridization buffer. Then samples were washed three times with PBS (1x) for 10 min each.Gelation was performed on the lid of a 4-well cell culture plate put on ice and covered with parafilm that served as a flat hydrophobic gelation surface. 18 mm cover glasses with cells facing down were placed on top of 90 μl pre-chilled ExM monomer solution (8.625% (w/w) SA, 20% (w/w) AA, 0.15% (w/w) BIS, 2 M NaCl in PBS) supplemented with 0.2% APS and 0.2% TEMED. Samples were then carefully transferred to a humidified chamber and incubated for 1.5 h at 37 °C for chemical crosslinking of acrylic monomers and trifunctional labels. After gelation samples were treated with 8 U Proteinase K in digestion buffer (50 mM Tris (pH 8.0), 0.5% TritonX-100, 0.8 M guanidine hydrochloride, 1 mM EDTA) and then expanded in double-deionized water. Water was exchanged several times until the maximum expansion factor of the hydrogel was reached. The expansion factor was determined by measuring the diameter of the gel using a calipser. When the expansion factor did not change within three water exchanges this factor was assumed as maximum expansion of the hydrogel.

**Protein Retention protocol (proExM protocol)**. Blocking and immunostaining were performed as described under "Immunostaining of unexpanded Cos-7 cells" incubating anti-α-tubulin antibody (ab1825, diluted 1:500, $c_{end} = 2$ μg/ml) and anti-ß-tubulin (T8328, diluted 1:200, $c_{end} = 10$ μg/ml) simultaneously in blocking buffer as primary antibodies and Alexa Fluor 532 IgG antibodies (A-11002 and A-1109, each diluted 1:200 to $c_{end} = 10$ μg/ml) diluted in blocking buffer as secondary antibodies.For copolymerization of amine groups into the hydrogel, cells were treated with the amine reactive agent Acryloyl X-SE (0.1 mg/ml) in PBS. The agent was freshly prepared from desiccated stock aliquots kept at −20 °C, incubated overnight in a humidified chamber, and subsequently washed twice for 15 min each in PBS (1x). Hydrogel formation, Proteinase K digestion and expansion in water were performed as described under "ExM protocol". After re-embedding of expanded hydrogels as described in section "Bind-silane treatment and re-embedding", samples were labelled with α-tubulin primary antibody solution (ab1825, diluted 1:500, $c_{end} = 2$ μg/ml) in 2% BSA for 3 h at 37 °C and then washed twice with 0.01% Tween in PBS for 20 min each and twice in PBS (1x) for 10 min each. Secondary antibodies were incubated for 3 h at 37 °C and washed twice with 0.01% Tween in PBS for 30 min each and twice with PBS (1x) for 30 min followed by a washing step over night in PBS (1x).

**ExM protocol with glutaraldehyde linker (ExM-GA protocol)**. Blocking and immunostaining were performed as described under "Immunostaining of unexpanded Cos-7 cells" using α- (ab1825, diluted 1:500, $c_{end} = 2$ μg/ml) and ß-tubulin (T8328, diluted 1:200, $c_{end} = 10$ μg/ml) antibodies as primary antibodies and a mixture of Alexa Fluor 532 IgG secondary antibodies (A-11002 and A-1109, each diluted 1:200 to $c_{end} = 10$ μg/ml) in blocking buffer. After washing with PBS (1x), cells were incubated with 0.25% GA in PBS for 10 min and washed thrice in PBS (1x) for 5 min each before proceeding with gelation of the samples. Gelation, digestion, and expansion was performed as described under "ExM protocol".

**DNA label with Cy5 (DNA-Cy5 protocol)**. Blocking and immunostaining were performed as described under "ExM protocol" with a mixture of primary α- and ß-tubulin antibodies (ab1825 diluted 1:500 with 2 μg/ml and T8328 diluted 1:200 with 10 μg/ml) and DNA conjugated secondary antibodies "Antibody B Cy5" or "Antibody C Cy5" in hybridization buffer that were then directly incorporated into the hydrogel. Hydrogel formation, proteinase K digestion and expansion were performed as described under "ExM protocol". After re-embedding on 24-mm silanized round coverslips samples were incubated over night with Cy5 antisense oligos with a DNA concentration of 0.5 ng/μl for each oligo in hybridization buffer.

**Ultrastructure expansion microscopy (U-ExM)**. Twelve millimeters cover glasses with isolated[2] centrioles were placed in a solution containing 0.7% FA, 1% AA diluted in PBS (1x). Next, 35 μl of pre-chilled U-ExM monomer solution (19% (w/w) SA, 10% (w/w) AA, 0.1% (w/w) BIS) supplemented with 0.5% APS and 0.5% TEMED in PBS for 1 min on a parafilm coated plate put on ice. Gelation proceeded for 1 h at 37 °C in a humidified chamber. Samples were placed in denaturation buffer (200 mM SDS, 200 mM NaCl in 50 mM Tris (pH 9.0)) for 15 min and then gels were carefully removed from the cover glasses and transferred to 1.5 ml centrifuge tubes filled with denaturation buffer. Hydrogels were then incubated for 30 min at 95 °C and then expanded in double deionized water until the maximum expansion of the gels were reached. After re-embedding on Bind-silane treated 24-mm cover glasses, centrioles were labelled with anti alpha-tubulin primary antibodies (ab1825, $c_{end} = 2$ μg/ml) diluted 1:500 in 2% BSA in PBS for 3 h at 37 °C, washed twice with 0.01% Tween in PBS for 20 min each and twice with PBS (1x) for 10 min each. Next, secondary Alexa Fluor 647 F(ab')2 antibodies (A-21246,

1:200, $c_{end} = 10$ µg/ml) diluted in 2% BSA were incubated 3 h at 37 °C followed by two washing steps in 0.01% Tween in PBS for 30 min each and two washes with PBS (1x) for 30 min. Before imaging gels were washed once more overnight in PBS (1x). For imaging of unexpanded centrioles the primary antibody anti poly-glutamylated tubulin (Adipogen, 1:500) was diluted in 5% BSA in PBS and incubated for 1 h at room temperature, washed thrice in PBS for 5 min each, followed by incubation with secondary Alexa Fluor 647 F(ab')2 antibodies (A-21246, 1:200, $c_{end} = 10$ µg/ml) diluted in 2% BSA for 1 h. The samples were then washed twice in 0.01% Tween in PBS and once in PBS for 10 min each.

**Re-embedding of expanded hydrogels (Re-embedding protocol).** To avoid shrinking caused by $d$STORM photoswitching buffer and to prevent drifting of the hydrogel during image acqustion an uncharged acrylamide gel was crosslinked throughout the hydrogel while chemically binding it on Bind-silane treated cover glasses. Round 24-mm cover glasses (high precision) were sonicated successively in double-deionized water, absolute ethanol and 5 M potassium hydroxide for 20 min each and washed with deionized water between every sonication step and finally oven dried at 100 °C. 200 µl of Bind-silane working solution (5 µl Bind-Silane in 8 ml absolute ethanol, 200 µl glacial acetic acid, 1.8 ml double deionized water) were distributed evenly on cleaned 24-mm cover glasses and left for around 1 h until the solution was fully evaporated. Cover glasses were then rinsed with doubly deio-nized water and air-dried. Glasses were prepared shortly before use. For re-embedding expanded hydrogels were placed in 6-well cell culture plates and each sample was covered with 3 ml of freshly prepared Re-embedding solution (10% acrylamide, 0.15% *bis*-acrylamide, 0.05% APS, 0.05% TEMED in 5 mM Tris (pH 8.9)). Samples were incubated on a platform shaker twice with freshly prepared solution for 30 min each. Shaking of the Re-embedding solution is crucial in this step as it brings oxygen into the solution that prevents it from gelling to early. The stirring speed should be adjusted so that the liquid is in motion but the gels are not damaged. After the second incubation, samples were transferred on silanized coverglasses while carefully removing excess solution from the hydrogels using laboratory wipes. Another coverglass that was not silanized was placed on top of the hydrogels during the following steps. The whole setup was transferred to a humidified container equipped with gas injection holes. To accelerate gelation oxygen was extracted from the container by purging the chamber with nitrogen for 15 min. The samples were then incubated at 37 °C for 2 h. After polymerization of the re-embedding gel samples were washed at least thrice for 30 min in double deionized water. Coverglasses on top of the hydrogel come off themselves during washing or can be detached carefully after the first washing steps. Re-embedded gels were then placed in imaging buffer or staining buffer depending on the sub-sequent protocol.

**Microscopes.** Single-molecule localization microscopy (SMLM) image acquisition was performed on a custom-built setup with an inverted Zeiss Axio Observer Z1 (Carl Zeiss Microscopy) microscope equipped with a Definite Focus autofocusing system. For excitation of different fluorescent molecules the setup provides three iBeam smart diode lasers with 405 nm (100 mW output power), 488 nm (200 mW output power) and 640 nm (150 mW output power) and a DPSS (diode pumped solid state) 532 nm laser (gem532, Laserquantum). Lasers were filtered with laser clean-up filters according to the specific wavelength and focused on the back focal plane of the objective to achieve a wide field illumination. To match the aqueous refractive index of expanded samples a water-immersion objective (LD C-Apochromat 63x/1.15 W Corr M27, Carl Zeiss Microscopy) is implemented in the microscope. The excitation light passes a quad-band dichroic beam splitter (Di01-R405/488/532/635-25×36, BrightLine) combined with a quad-band rejection filter (ZET405/488/532/642 m, Chroma). For recording the emission of excited fluor-ophores the setup is equipped with two Andor Ixon Ultra 897 EMCCD (electron-multiplying charge-coupled device) cameras at the side port of the microscope. The software Andor Solis (Version 4.28.30014) was used to control the EMCCD cameras. The fluorescence light is parallelized through a 160 mm achromatic lens (Thorlabs) and can be spectrally separated by a 630 DCXR (Chroma) dichroic beam splitter. In this configuration, two different emission wavelengths can be focussed on two cameras arranged perpendicular to each other. For all $d$STORM measurement in this work the beam splitter was removed and the emission light was directed to one camera. Suitable emission light filters were placed in front of the camera depending on the detected fluorescent wavelength. For 3D imaging, an additional achromatic cylindrical lens ($f = 250$ mm, Thorlabs) was placed in the detection path close to the imaging plane before the relay system. Rescanning confocal imaging (RCM) was performed on a Nikon TiE inverted microscope equipped with an RCM unit (Confocal.nl) that is based on the image scanning principle[42]. The setup was operated by the microscope software NIS-Elements (version 4.6).

**Mounting and SMLM image conditions.** Re-embedded hydrogels immobilized on 24-mm cover glasses were immersed in photoswitching buffer consisting of 100 mM cysteamine hydrochloride (MEA) in PBS with optimized pH (adjusted with KOH) depending on the utilized fluorescent dye. For Alexa Fluor 647 and Cy5 fluorophores, the pH of the imaging buffer was adjusted to pH 7.7 and to pH 7.9 when using Alexa Fluor 532, respectively. The buffer was prepared freshly before

use. The hydrogel was incubated in photoswitching buffer twice for 20 min each before imaging. U-ExM treated samples labeled with the spontaneously blinking Si-rhodamine dye HMSiR were immobilized on Poly-L-lysine (0.1%) coated 24-mm high-precision cover glasses and additionally embedded in 1% (w/v) Agarose. As imaging buffer, double deionized water or pH adjusted PBS buffer (1x, pH 7.4) was used. For unexpanded $d$STORM imaging samples were placed in 100 mM MEA in PBS adjusted to pH 7.5 (with KOH) for DNA-Cy5 and Streptavidin-Alexa Fluor 647 and pH 7.9 (with KOH) when using Alexa Fluor 532.

**3D $d$STORM calibration.** To obtain 3D calibration curves, fluorescent beads were mixed in U-ExM or ExM monomer solution for 3D image acquisitions of U-ExM or ExM samples, respectively. Therefor fluorescent marker stock suspension (0.1 µm, ~1.8 × 10$^{11}$ particles/mL, TetraSpeck Microspheres, Thermo Fisher) was vortexed for ~1 min and then diluted 1:50 in the corresponding monomer solution. After adding TEMED and APS in the appropriate concentrations, the bead-gel solution was vortexed again for ~ 20 s, polymerized, and expanded as described under the respective expansion protocol (omitting the digestion or denaturation step). The expanded gels were then transferred on poly-L-lysine (0.1%) coated coverslips and additionally embedded in 1% (w/v) Agarose. 4 µm z-stacks of several fluorescent markers dispersed in the hydrogel ~50–400 µm above the coverslip were recorded and used to generate 3D calibration curves as described below. The software Micro-manager 1.4 was used for image acquisition and to control the piezo driven stage.

**Image processing.** For 2D and 3D $d$STORM image reconstruction super-resolution images were analyzed, post-processed and visualized using the analysis platform SMAP (Superresolution Microscopy Analysis Platform) with the GPU based 3D fitter fit3Dcspline[27] and the ImageJ plugin ThunderSTORM[43]. The respective integrated calibration tools were used for generating 3D astigmatism calibration curves. Localizations were further corrected for drift using the cross-correlation method, filtered for molecules with poor precision and grouped to one localization when molecules appeared in several consecutive images.

**Expansion factor determination.** Centriole diameters of U-ExM expanded sam-ples were determined by averaging peak-to-peak distances of two cross-sectional profiles that were drawn through the center of the ninefold-symmetrical α-tubulin signal using the line profile tool of Fiji[44]. Peaks were then determined by using the peakfinder minitool implemented in the analyse software Origin (OriginLab, Northampton, MA). To determine the expansion factor post-expansion and post-re-embedding, Cos-7 cells were labeled with a-tubulin and ß-tubululin and expanded according to the "proExM protocol". An additional post-expansion immunostaining for α-tubulin was performed using the same primary and sec-ondary antibodies. RCM images of the same cells were acquired before gelation, after expansion and after re-embedding in different imaging buffers. Images were then registered via rigid (similarity) and non-rigid registration (B-spline) using the open source, command-line program elastix[6]. The transform parameters of the similarity transformation of pre- and post-expansion RCM images were used to determine the initial expansion factor of the sample. RCM images acquired post-re-embedding in PBS (1x) and cysteamine hydrochloride as well as a $d$STORM image in photoswitching buffer of the same area were registered in the same way using elastix to determine the expansion factor after re-embedding in different imaging buffers. Furthermore, a deformation vector field of pre-expansion and post re-embedding RCM images was created using elastix and transformix[6]. Elastix and transformix code were executed in Wolfram Mathematica 11.2.

**Analysis of microtubule transversal profiles.** To analyze and compare the dif-ferent expansion protocols we developed a home written software that detects fiber like structures and automatically determines the transversal profile along these structures in reconstructed SMLM images. In detail the SMLM images are first convolved with a Gaussian blur compensating for noise discontinuity or holes. A thresholding algorithm[45] then converts the image from grayscale to binary. Using Lees algorithm[46] the expanded lines are reduced to one pixel width. The pixel coordinates from all still connected lines are then retrieved and tested for continuity. Points of discontinuity are used as breakpoints and all following coordinates are connected to a new line. Lines, shorter than the minimum required length are discarded. An univariate spline of degree 3 (c-spline) is fitted to each line. Note that shape and gradient of the line depend on the smoothing parameter. The result is a table containing the spline coordinates and the local derivatives. Perpendicular to the derivative a line profile is extracted from the original image at each coordinate point. The averaged profiles for each spline are fitted with the following functions (Eqs. (1–5)):

$$\text{Gaussian}: \quad y = h\mathrm{e}^{\frac{-(x-c)^2}{2w^2}} + b \qquad (1)$$

(where $h$ is the intensity, $c$ the center, $b$ the offset, and $w$ the variance of the distribution. Optimal for single profiles).

$$\text{Bi-Gaussian}: \quad y = h_1\mathrm{e}^{\frac{-(x-c_1)^2}{2w_1^2}} + h_2\mathrm{e}^{\frac{-(x-c_2)^2}{2w_2^2}} + b \qquad (2)$$

(optimal for profiles containing a dip).

$$\text{Tri}-\text{Gaussian}: y = h_1 e^{\frac{-(x-c_1)^2}{2w_1^2}} + h_2 e^{\frac{-(x-c_2)^2}{2w_2^2}} + h_3 e^{\frac{-(x-c_3)^2}{2w_3^2}} + b \qquad (3)$$

(optimal for profiles exhibiting a dip and high background signal).

$$\text{Cylinder}: y = \begin{cases} h\left(\sqrt{r_2^2 - (x-c)^2} - \sqrt{r_1^2 - (x-c)^2}\right), \text{if } \|x\| < r_1 \\ h\left(\sqrt{r_2^2 - (x-c)^2}\right), \text{if } \|x\| \ge r1, \|x\| < r_2 \\ 0, \text{else} \end{cases} \qquad (4)$$

(y describes the theoretical intensity profile of microtubules where $r_1$ and $r_2$ denote the inner and outer cylinder radius. The quality of the fit strongly depends on the initial estimation of the parameters, due to the nonlinearity of the cylinder function.)

$$\text{Multi}-\text{Cylinder}: y = cyl(i_1, c, 25e_x/2 - 2a, 25e_x/2 - a) + cyl$$
$$(i_2, c, 42.5e_x/2, 42.5e_x/2 + a) + cyl(i_3, c, 25e_x/2 + a, 25e_x/2 + 2a) + b \qquad (5)$$

(includes the theoretical dimensions of microtubules leaving less degrees of freedom. Might result in a better fit). Note that the fit intensity ($h$) gives a good estimation for the relative labeling density.

Using the splines fitted to the maximum intensity projection we constructed $xz$-profile projections of microtubules, by taking line profiles in each $z$-stack of the 3D image. Averaging the aligned line profiles in a layer yields the intensity values for the corresponding row of the $xz$-projection.

**Reporting summary**. Further information on research design is available in the Nature Research Reporting Summary linked to this article.

## Data availability
All data that support the findings described in this study are available within the manuscript, the related supplementary information or deposited at https://doi.org/10.6084/m9.figshare.12415787.v1. Additional information is available from the corresponding authors upon reasonable request.

## Code availability
The automated image processing software Line Profiler is available at https://line-profiler.readthedocs.io/en/latest/.

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

## Acknowledgements

The authors thank P. Gessner and L. Behringer-Pliess for assistance in immunocytochemistry and cell culture preparation. This work was supported by the German Research Foundation (DFG, TRR 166 ReceptorLight, project A04) and the European Regional Development Fund (EFRE project "Center for Personalized Molecular Immunotherapy"). This work is supported by the European Research Council (ERC; StG 715289 ACCENT to P.G.) and the Swiss National Science Foundation (SNSF) PP00P3_187198 to P.G.

## Author contributions

F.U.Z., S.R., D.G., T.D.M.B., V.H., P.G., and M.S. conceived and designed the project. M.S, V.H., and P.G supervised the project. F.U.Z. performed all Ex-SMLM experiments. S.R. developed Line Profiler and analyzed the data together with F.U.Z., D.G. provided the centriole samples. All authors wrote and revised the final manuscript.

## Competing interests

The authors declare no competing interests.
