## [Peer Review File · Nature Communications]

Reviewers' Comments:

Reviewer #1:

Remarks to the Author:

In this work, Zwettler et al. demonstrated the stochastic optical reconstruction microscopy (STORM) imaging of expanded specimens and quantitatively analyzed the images. Recently, multiple papers have demonstrated the super-resolution microscopy imaging of expanded specimens to achieve an even higher resolution. In comparison with those papers, this paper demonstrates the highest lateral and axial resolution and provides a very detailed quantitative analysis. I believe that this paper will be very useful to super-resolution imaging societies because it not only demonstrates STORM imaging of expanded specimens but also provides a detailed comparison of different ExM labeling procedures. The authors have addressed all of my comments, and I do not have any objections to the publication of this paper in *Nature Communications*.

Reviewer #2:

Remarks to the Author:

This is work from one of the leading laboratories in single molecule localisation-based super resolution microscopy in collaboration with a group working on the centriole.

The authors develop an Expansion microscopy scheme, in which Expansion microscopy can be combined with single molecule localisation microscopy. This is of high importance, as so far Expansion microscopy has been successfully combined with structured illumination and STED microscopy, but efforts to combine ExM with SMLM, which arguably has highest possible resolution of the widely used superresolution techniques, has not yielded convincing results. This is very exciting as the combination of ExM with SMLM could lead to a resolution of structures in the range of few nanometers.

The authors show very high quality data for their method in 2D and 3D on microtubules and on centrioles.

The trick the authors use to achieve combination of ExM with SMLM is that they generate a second gel after the first gel is expanded and crosslink it to the expanded first gel. Now the first gel, which was expanded in deionized water cannot drink again when the higher osmolarity blinking buffer is added. This is nicely demonstrated by exchanging the most commonly used AF647 dye, which requires blinking buffer for HMSiR, which blinks (mostly) independently of the buffer. When this dye is used, the same expansion is achieved for imaging in deionized water, but when this is exchanged against PBS, the gel shows a smaller expansion factor.

Overall, the work is important and convincing and I support publication after minor changes to the manuscript have been made.

Minor points: The sentence in the legend to Figure 4: "(d) Comparison of diameter of expanded centrioles re-embedded and labeled with Alexa Fluor 647 imaged with dSTORM in MEA photoswitching buffer labeled with HMSiR 647 imaged in double-deionized water or in pH optimized PBS (1x) buffer with pH 7.4." does not seem to make sense.

Point-by-point response to the referees' comments:

We wish to thank the reviewers for their positive and constructive comments that have helped to improve the manuscript and strengthen its novelty, performances and the fields of applications. We have revised the manuscript in depth taking into account all the remarks.

Reviewer #1 (Remarks to the Author):

In this work, Zwettler et al. demonstrated the stochastic optical reconstruction microscopy (STORM) imaging of expanded specimens and quantitatively analyzed the images. Recently, multiple papers have demonstrated the super-resolution microscopy imaging of expanded specimens to achieve an even higher resolution. In comparison with those papers, this paper demonstrates the highest lateral and axial resolution and provides a very detailed quantitative analysis. I believe that this paper will be very useful to super-resolution imaging societies because it not only demonstrates STORM imaging of expanded specimens but also provides a detailed comparison of different ExM labeling procedures. The authors have addressed all of my comments, and I do not have any objections to the publication of this paper in Nature Communications.

We thank the reviewer for his/her positive valuation.

Reviewer #2 (Remarks to the Author):

This is work from one of the leading laboratories in single molecule localisation-based super resolution microscopy in collaboration with a group working on the centriole. The authors develop an Expansion microscopy scheme, in which Expansion microscopy can be combined with single molecule localisation microscopy. This is of high importance, as so far Expansion microscopy has been successfully combined with structured illumination and STED microscopy, but efforts to combine ExM with SMLM, which arguably has highest possible resolution of the widely used superresolution techniques, has not yielded convincing results. This is very exciting as the combination of ExM with SMLM could lead to a resolution of structures in the range of few nanometers.

The authors show very high quality data for their method in 2D and 3D on microtubules and on centrioles.

The trick the authors use to achieve combination of ExM with SMLM is that they generate a second gel after the first gel is expanded and crosslink it to the expanded first gel. Now the first gel, which was expanded in deionized water cannot drink again when the higher osmolarity blinking buffer is added. This is nicely demonstrated by exchanging the most commonly used AF647 dye, which requires blinking buffer for HMSiR, which blinks (mostly) independently of the buffer. When this dye is used, the same expansion is achieved for imaging in deionized water, but when this is exchanged against PBS, the gel shows a smaller expansion factor.

Overall, the work is important and convincing and I support publication after minor changes to the manuscript have been made.

Minor points: The sentence in the legend to Figure 4: "(d) Comparison of diameter of expanded centrioles re-embedded and labeled with Alexa Fluor 647 imaged with dSTORM in MEA photoswitching buffer labeled with HMSiR 647 imaged in double-deionized water or in pH optimized PBS (1x) buffer with pH 7.4." does not seem to make sense.

We thank the reviewer for his/her valuable comments and careful reading. We corrected the legend of Figure 4. It reads now: “**d**, Comparison of the diameters determined from expanded centrioles measured using different protocols (re-embedded and labeled with Alexa Fluor 647, and imaged in MEA photoswitching buffer, labeled with HMSiR 647 and imaged in double-deionized water or in pH optimized PBS (1x) buffer with pH 7.4).”